# With or without You: Co-Chaperones Mediate Health and Disease by Modifying Chaperone Function and Protein Triage

**DOI:** 10.3390/cells10113121

**Published:** 2021-11-11

**Authors:** Selin Altinok, Rebekah Sanchez-Hodge, Mariah Stewart, Kaitlan Smith, Jonathan C. Schisler

**Affiliations:** Computational Medicine Program, Department of Pharmacology, Department of Pathology and Lab Medicine, McAllister Heart Institute, The University of North Carolina at Chapel Hill, Chapel Hill, NC 27599, USA; saltinok@email.unc.edu (S.A.); buckeyebecky@unc.edu (R.S.-H.); mariahss@live.unc.edu (M.S.); kaitlans@live.unc.edu (K.S.)

**Keywords:** heat shock proteins, co-chaperones, protein quality control, protein folding, protein degradation, cardioprotection, neuroprotection, cancer

## Abstract

Heat shock proteins (HSPs) are a family of molecular chaperones that regulate essential protein refolding and triage decisions to maintain protein homeostasis. Numerous co-chaperone proteins directly interact and modify the function of HSPs, and these interactions impact the outcome of protein triage, impacting everything from structural proteins to cell signaling mediators. The chaperone/co-chaperone machinery protects against various stressors to ensure cellular function in the face of stress. However, coding mutations, expression changes, and post-translational modifications of the chaperone/co-chaperone machinery can alter the cellular stress response. Importantly, these dysfunctions appear to contribute to numerous human diseases. Therapeutic targeting of chaperones is an attractive but challenging approach due to the vast functions of HSPs, likely contributing to the off-target effects of these therapies. Current efforts focus on targeting co-chaperones to develop precise treatments for numerous diseases caused by defects in protein quality control. This review focuses on the recent developments regarding selected HSP70/HSP90 co-chaperones, with a concentration on cardioprotection, neuroprotection, cancer, and autoimmune diseases. We also discuss therapeutic approaches that highlight both the utility and challenges of targeting co-chaperones.

## 1. Introduction

### 1.1. Protein Quality Control

Heat shock proteins, such as HSP70 and HSP90, protect cellular homeostasis and play a vital role in responding to multiple forms of cell stress (Figure 1). As molecular chaperones of the protein quality control (PQC) machinery, HSPs exert their protective function in various ways: (1) facilitating the folding of nascent proteins into their native state; (2) enabling the formation of multiprotein complexes; (3) refolding stress-damaged, misfolded proteins; and (4) promoting the degradation of misfolded or aggregated proteins by linking the PQC machinery to the ubiquitin-proteasome system or the autophagy-lysosome system [1,2]. However, a network of proteins called co-chaperones modifies HSP function. Via protein–protein interactions, co-chaperones alter the refolding activity of HSPs and expand an HSP’s client protein portfolio via facilitating additional interactions between chaperones and their substrates [2,3]. Disruptions of the chaperone/co-chaperone machinery often result in the accumulation of misfolded or aggregated proteins that lead to proteotoxicity. Coding mutations in numerous co-chaperones are associated with multiple human diseases [4,5,6,7]. HSPs and their co-chaperones have been extensively studied in different disease settings, including but not limited to cardiovascular diseases, cancer, and neurodegenerative diseases. Numerous studies on co-chaperones over the past twenty years detail how these proteins alter chaperone function. However, we are only beginning to understand how the co-chaperone network coordinates with heat shock proteins and the cellular degradative machinery to regulate protein quality control. In this review, we take a disease-centric approach and describe the results of studies on several co-chaperones, how they modify chaperone function, possible mechanisms of disease biology, and therapeutic considerations.

### 1.2. Cardiac Stress

Due to constant contractility, the human heart produces and uses significant amounts of ATP, around 100 times more than its weight, within 24 h [8,9]. This high metabolic activity of the heart, particularly the cardiomyocytes (CM), creates a strict demand for an efficient PQC system to ensure proper protein synthesis, folding, and degradation [10]. Many heart diseases stem from the accumulation of misfolded proteins due to mechanic stress, oxidative stress, and pH changes; regardless of the underlying conditions that cause heart disease, such as heart failure (HF), myocardial infarction (MI), or genetic mutations [7,11,12]. HSPs play a central role in cardiac PQC, most notably in response to stress [7,11,12]. HSPs provide cardioprotection by preventing the accumulation of misfolded proteins, inhibiting myocardial cell death pathways, regulating ion channels, and impeding the function of pro-inflammatory cytokines [13]. Moreover, the importance of HSPs in heart function is highlighted in studies showing the protective effects of HSP-inducing therapeutics during cardiovascular diseases [14,15,16,17,18].

### 1.3. Neurodegeneration

Like cardiomyocytes, neurons are particularly susceptible to protein aggregate-mediated proteotoxicity due to their post-mitotic state and highly intricate structure [19,20,21]. HSPs are vital in ensuring neuronal homeostasis by promoting the clearance of aggregated proteins by the ubiquitin-proteasome system or the autophagy-lysosome system [19]. However, when targeting misfolded proteins to either chaperone-mediated refolding or degradative pathways fail, protein aggregates form and disrupt critical neuronal processes, eventually resulting in neuronal death [20]. The accumulation of aggregated proteins is one of the main features of neurodegenerative diseases, including Alzheimer’s Disease (AD), Parkinson’s Disease (PD), polyglutamine diseases, and several spinocerebellar ataxias (Figure 2). The disease-causing protein aggregates in these conditions are briefly covered below, highlighting the importance of HSP-dependent aggregate clearance to maintain neuronal function. 

Alzheimer’s Disease is a progressive neurodegenerative disease with two main characteristic features: hyperphosphorylated tau protein and amyloid-β peptide accumulation in neurofibrillary tangles and amyloid plaques, respectively [22]. The hallmark of Parkinson’s Disease is the formation of Lewy bodies, a buildup of aggregated α-synuclein. Genetic mutations cause familial forms of PD, including α-synuclein (*SNCA*), parkin (*PRKN*), leucine-rich repeat kinase-2 (*LRRK2*), and PTEN-induced putative kinase 1 (*PINK1*), which all individually result in aberrant α-synuclein activity and its subsequent aggregation [23,24,25]. Polyglutamine (PolyQ) diseases are other examples of neurodegenerative diseases caused by protein aggregates. Cytosine-adenine-guanine (CAG) trinucleotide repeat expansion mutations in disease-related genes result in long stretches of glutamine residues (PolyQ) when translated [26,27]. PolyQ protein buildup eventually leads to their aggregation, disrupting neuronal processes and consequent neuronal death [27]. Huntington’s Disease (HD) is a hereditary autosomal dominant disease caused by CAG repeat expansion in the Huntingtin gene, *HTT*, which encodes huntingtin protein (htt). These CAG repeat expansions translate into a long glutamine chain attachment to huntingtin protein, and as a result, PolyQ-htt aggregates in inclusion bodies, causing neurotoxicity [28]. Patients diagnosed with spinocerebellar ataxias suffer from impaired balance or coordination, gait abnormalities, speech disruptions, and irregularities in eye movement due to cerebellar degeneration [29]. Disease-causing mutations span different genes and present as either autosomal dominant or recessive forms of spinocerebellar ataxias. Out of all identified autosomal dominant spinocerebellar ataxias (SCAs), SCA types 1, 2, 6, 7, and 17 are identified as PolyQ diseases since the mutations in their causative genes result in PolyQ expansion which alters the proteins’ function [30]. 

### 1.4. Cancer

HSP expression is upregulated in various cancers in which HSPs promote tumor initiation, metastasis, and treatment resistance [6]. HSPs exert these functions by facilitating multiple different hallmarks of cancer, such as sustained growth, evasion of cell death, and resisting growth suppressors [31]. HSPs can inhibit the function of a primary tumor suppressor, p53, promoting cancer cell growth [32,33]. Moreover, HSP-mediated inhibition of pro-apoptotic pathways provides further advantages to cancer cells to survive [34,35]. Since cancer cells rely on HSPs to survive, HSP inhibitors gained importance as a possible cancer treatment. 

Several well-written reviews describe HSP90 inhibitors in cancer [36,37,38]. Most recognizable in the literature is 17-AAG, an inhibitor that binds to HSP90’s ATP binding site. 17-AAG decreased cancer proliferation and growth in breast, colorectal, and head and neck cancers [39,40,41]. In a panel of tumor types including liver, brain, kidney, lung, and prostate, pan-HSP90 inhibitors targeted to the N-terminus, both alone and in combination, induced mitochondrial dysfunction and cytotoxicity, resulting in decreased tumor size and cell proliferation in vitro and in vivo [42]. This panel highlighted the broad use of HSP90 inhibitors in different cancer types. A more extensive study in thyroid cancer cell lines confirmed HSP90 inhibitors also targeted to the N-terminus decreased cancer cell migration and invasion [43]. Another study by Wei et al. showed the anticancer effect of inducing apoptosis and impairing autophagic flux in the lung cancer cell line A549 using a derived inhibitor that binds to the C-terminus of HSP90 [44]. However, drugs such as ganetespib, capecitabine, tanespinmycin, and irinotecan (known HSP90 inhibitors) had modest anticancer effects in clinical trials for the treatment of colorectal cancer [45]. Additionally, the expression and activity of HSPs increase in response to numerous cell stress events (Figure 1). As such, HSP inhibitors could impact the stress response in normal cells that rely on HSPs for survival versus cancerous cells. 

### 1.5. Autoimmune Disorders

In recent years, several studies identified the roles of chaperones and co-chaperones in regulating inflammatory responses and how dysfunction of the chaperone system related to these responses leads to autoimmune diseases [46,47]. Exciting new roles for chaperones and co-chaperones related to inflammation, autoimmune disorders, and the previously described diseases include mitochondrial and nuclear shuttling of cargo and chaperone activity within these organelles (Figure 1). We encourage the reader to consider several reviews that cover chaperone-organelle specific activities and their role in disease in greater detail [48,49,50,51,52,53,54,55,56].

## 2. CHIP

### 2.1. Function, Expression, and Regulation

CHIP, Carboxyl terminus of HSC70 Interacting Protein, was discovered in a human heart cDNA screen for TPR-domain proteins and subsequently confirmed to be a binding partner of HSC/HSP70 [57]. There are three functional domains of CHIP, each contributing to the multiple activities attributed to this enzyme: (1) the TPR domain, mediating chaperone interactions, (2) a coiled-coil domain, essential for dimerization, and (3) the U-box, necessary for E3 ubiquitin ligase activity [58,59]. Together, these activities allow CHIP to engage chaperones and facilitate ubiquitination of chaperone-bound clients, positioning CHIP as a critical mediator for PQC [60,61]. 

The TPR domain of CHIP binds directly with the C-terminal EEVD motifs found in HSC/HSP70 and HSP90. The binding affinity of CHIP towards HSP90 is 6-fold higher than HSP70 in vitro [62]; however, given the concentrations of these proteins and other competing co-chaperones in cells, CHIP is likely complexed with HSC/HSP70 more than HSP90 [62,63]. The impact of other TPR domain-containing proteins on CHIP function and chaperone output is crucial in understanding protein quality control. The co-chaperone HSP70-HSP90 Organizing Protein (HOP) directly competes with CHIP for chaperone binding [57,63,64]. HOP is not a ubiquitin ligase and, therefore, triage outcomes of chaperone substrates, refolding versus degradation, are impacted based on the binding dynamics of these co-chaperones. Most CHIP studies, including our own, overlook how manipulations affect factors such as HOP. However, several studies make it clear that looking at the more extensive chaperone/co-chaperone system is critical in our understanding of the underlying biology [62,63,64,65,66,67]. 

Both HSC/HSP70 and HSP90 utilize additional co-factors to promote refolding, such as HSP40 and AHA1, respectively. Nucleotide exchange for HSP70 is the rate-limiting factor for chaperones to release their substrate, and the presence of other co-chaperones can impact the ATPase activity of these chaperone complexes [68]. CHIP inhibits the ATPase activity of the HSC/HSP70 and HSP40, limiting client refolding [57,62]. In contrast, ATPase activity is not changed when CHIP engages the HSP90/AHA1 complex [62,63]; however, the rate-limiting step for HSP90 is a conformational change from the open to closed state, an event that precedes ATP hydrolysis [69]. Overall, these data implicate CHIP in limiting HSC/HSP70 substrate refolding and promoting ubiquitination of misfolded HSC/HSP70 substrates to target them for degradation. The physiological role of CHIP and HSP90-bound substrates is less clear, but since several HSP90 clients are regulatory proteins, CHIP and ubiquitin signaling may play a cell signaling role.

Subsequent studies found that CHIP co-localizes and interacts with proteasome subunits [70,71,72], reinforcing CHIP’s role in chaperone-mediated protein triage. CHIP also ubiquitinates HSC/HSP70 and HSP90, and this regulatory role may be necessary for attenuating the heat shock response [73,74]. CHIP is expressed throughout all tissue types and higher in metabolically active tissues such as the heart, brain, and muscle, suggesting an increased dependence of CHIP in these systems [57]. Several reviews provide detailed information on CHIP and its role in neurological diseases, tumorigenesis, heart failure, and immunity [75,76,77,78]. The regulation of CHIP expression and function is still an emerging field of study; however, post-translational modifications appear to play a vital role, including mono- and auto-ubiquitination [79] and phosphorylation [75]. Understanding CHIP regulation at the genetic to the post-translational level undoubtedly is a critical gap in the field. 

### 2.2. Cardioprotection

Several labs identified cardioprotective roles for CHIP (Figure 3). CHIP protects cardiomyocytes by promoting anti-inflammatory and anti-oxidative processes, suppressing the activation of NF-kB and P38 MAPK, and down-regulating pro-apoptotic proteins such as caspase-3 and Bax [80]. Silencing CHIP in rat neonatal cardiomyocytes exacerbated reactive oxygen species and inflammation in hyperglycemic conditions [80]. Loss of CHIP expression in mouse models increased susceptibility to ischemia-reperfusion injury [81] and decreased fractional shortening, increased mortality, and accelerated left ventricular hypertrophy in response to cardiac pressure overload via trans-aortic banding (Figure 3) [82]. Schisler et al. confirmed a previous in vitro study [83] that CHIP can function as an autonomous chaperone, and in the heart, CHIP regulates cardiac metabolism by chaperoning AMPK [82]. Nuclear receptors, including PPARα, PPARγ, and ERRα, are well-characterized clients of HSP70 and HSP90. PPARγ and ERRα are ubiquitinated by CHIP leading to proteasomal degradation [84,85]. CHIP also inhibits PPARβ transcriptional activity, although the mechanism is not understood [86]. Recently, our lab found that fenofibrate, a PPARα agonist and lipid-lowering drug, caused cardiac fibrosis and reduced cardiac function in mice lacking CHIP expression [87]. These studies suggest that the co-chaperone activity of CHIP contributes to the regulation of nuclear receptors. 

Whereas loss-of-function studies highlight a critical role for CHIP in the heart in response to stress, increased CHIP expression, or a gain-of-function engineered version of CHIP confers cardioprotection. Overexpressing CHIP in vivo via cardiomyocyte-specific transgenics conferred protection against pathological remodeling and prevented loss of function after myocardial infarction (Figure 3) [88,89]. CHIP-mediated cardioprotection resulted in new blood vessels in hearts after myocardial infarction, decreased expression of p53, MCP-1, and ICAM-1 (Figure 3), reduced proinflammatory cytokine expression, and macrophage infiltration [88,89]. Reduced heart inflammation was confirmed by the CHIP-dependent attenuation of NF-kB/p65, p38, and JNK activity (Figure 3) [88,89]. Ranek et al. identified CHIP-S20 as a target of protein kinase G, and phosphorylation of this serine, located in the TPR domain of CHIP, results in a prolonged half-life of CHIP protein [75]. Mice engineered with a phosphomimetic form of CHIP (CHIP-S20E) were protected against myocardial infarct, with lower mortality rates and decreased infarct size than wild-type mice [75]. Post MI, CHIP-S20E mouse hearts had less ubiquitinated proteins [75] suggesting the functionally enhanced version of CHIP promotes proteostasis.

Heart failure with a preserved ejection fraction (HFpEF) is a common and untreatable form of HF. A recent study suggests that CHIP is regulated directly by Xbp1s (a sliced form of the X-box-binding protein) and a potential therapeutic target for HFpEF through its ability to degrade the transcription factor FoxO1 [90]. In rat neonatal cardiomyocytes, CHIP expression alleviated myocardial lipid formation, and either the loss of FoxO1 or the over-expression of Xbp1s eliminates the HfpEF phenotype in mouse models [90]. In a complementary study, treating mice with Imeglimin restored cardiac CHIP expression, decreased FoxO1 levels, and decreased fatty acid synthase [91], demonstrating a protective role for CHIP against apoptosis and oxidative stress.

### 2.3. Neurodegenerative Diseases

#### 2.3.1. Parkinson’s Disease

CHIP also plays a protective role against several models of aggregated protein-mediated toxicity in the brain (Figure 2), recently reviewed by Zhang et al. [78]. In some diseases, the link to chaperones is clear. CHIP co-localizes with HSP70 and α-synuclein in Lewy bodies, abnormal protein aggregations involved in the neurotoxicity seen in Parkinson’s disease [58]. CHIP-mediated protection via overexpressing CHIP requires a functional TPR domain and HSP70 [58], highlighting how the co-chaperone function of CHIP could be targeted for PD therapies. Other mechanisms involving CHIP-chaperone interactions include inherited forms of PD: CHIP-HSP90-dependent degradation of leucine-rich repeat kinase-2 (*LRRK2*) [92,93], CHIP-HSP70-dependent enhancement of the E3 activity of Parkin (*PRKN*) [94], and CHIP-HSP70 dependent degradation of PTEN-induced putative kinase 1 (*PINK1*) [95]. It is compelling that at least three genes involved in familial Parkinson’s disease are regulated by CHIP and HSP70/90.

#### 2.3.2. Alzheimer’s Disease

The role of CHIP in Alzheimer’s disease is less clear. Tau is a microtubule-associated protein that binds and stabilizes the neuronal microtubule network. Upon hyper-phosphorylation and dissociation from microtubules, tau aggregates into neurofibrillary tangles to form a defining neuropathological lesion in AD, correlating with neurodegeneration and neuronal death (Figure 2) [96]. CHIP was initially implicated in direct ubiquitination and degradation of tau [97,98,99,100], consistent with the detection of tau within and surrounding proteasomes [101,102,103]. However, in complete contrast, several reports have proposed chaperone-dependent/ubiquitin-independent roles for CHIP and even suggested a paradoxically limited role for CHIP in tau degradation [104,105]. While CHIP can be found in AD brains, it is unknown whether CHIP plays an active role in neuroprotection against Alzheimer’s disease. Knocking out CHIP in a mouse model of AD resulted in a remarkable increase in hyper-phosphorylated tau levels; however, total tau levels were not appreciably changed, as might be expected in the absence of CHIP [97]. These data suggest that CHIP likely regulates tau directly via non-degradative mechanisms or indirect interactions [106,107]. CHIP inhibits tau accumulation by promoting ubiquitination and degradation of HDAC6, a known HSP90 modulator that regulates protein refolding or degradation decisions [106]. Interestingly, HDAC6 inhibition shifts the refolding-degradation balance towards degradation and results in HSP90 client tau protein degradation [106]. Another example of CHIP’s indirect regulation of tau is mediated through Akt-CHIP interactions [107]. Akt-mediated MARK2 activity enhancement results in tau phosphorylation at S2626/S345, which prevents tau’s recognition by CHIP, promoting tau aggregation [107]. 

#### 2.3.3. Spinocerebellar Ataxias

Soon after the characterization of CHIP, several studies linked CHIP dysfunction to ataxic phenotypes (Figure 2). However, it was nearly 15 years after the discovery of CHIP when the first disease-causing coding mutation in the gene that encodes CHIP (*STUB1*) was identified in a family with two siblings with early-onset ataxia [108,109]. Soon after the report of the homozygous mutation, resulting in a missense mutation in CHIP’s U-box (CHIP-T246M), numerous studies identified other mutations, and the disease was classified as Spinocerebellar Ataxia Autosomal Recessive 16 (SCAR16) [109]. Looking at the myriad of experimental and clinical reports from the past eight years [110], it is clear that SCAR16 is a neurodegenerative disease that displays a range of clinical phenotypes, including accelerated aging, cognitive dysfunction, ataxic gait, and hypogonadism [111,112]. A follow-up study from our group identified that the CHIP-T246M mutation results in a structural change to CHIP’s U-box domain, leading to loss of E3 ubiquitin ligase activity while increasing the interaction between mutant CHIP and HSC(P)70 [112]. CRISPR/Cas9 edited mice and rats harboring the CHIP-T246M mutation also exhibited age-dependent changes in gait and cognitive dysfunction, similar to the symptoms observed in SCAR16 patients [112]. 

Currently, over 30 SCAR16 disease-associated mutations occur in all three functional domains of CHIP [110]. Our lab examined the relationship between mutation locations, the changes in CHIP function, and the clinical phenotypes of SCAR16 patients [110]. Interestingly, U-box mutations are associated with cognitive dysfunction. In contrast, TPR mutations did not show this pattern [111], suggesting that the loss or gain of specific functions or CHIP may contribute to the heterogeneity in patient phenotypes. 

In 2019, heterozygous *STUB1* mutations identified in patients with ataxia uncovered a new classification of autosomal dominant spinocerebellar ataxia, SCA48 [113]. To date, 19 different CHIP mutations have been identified in SCA48 patients [110,114,115]. SCA48-associated disease mutations are limited to the TPR and U-box domain in CHIP with one exception; a nonsense mutation (p.R225*) at the end of the coiled-coil domain that results in the deletion of the entire U-box domain [110]. A recent study identified an increase in tau and α-synuclein aggregates in cells transfected with SCA48-associated CHIP-G278fs mutation [116]. While the disease-causing CHIP mutations of the dominant and recessive forms of spinocerebellar ataxias mostly differ, there are common mutations in both diseases. However, the mechanism behind the dominant heterozygous CHIP mutations that cause SCA48 is unknown. One compelling explanation is that these mutations mimic haploinsufficiency, which may explain the late onset of disease that has been seen in SCA48 patients [110].

#### 2.3.4. Polyglutamine Diseases

A study identified the link between polyQ diseases and CHIP, demonstrating that CHIP reduced the accumulation of insoluble protein aggregates, polyQ accumulation, and toxicity in primary neurons (Figure 2) [117]. The protective effects of CHIP required a functional TPR domain, indicating that interactions with chaperones are essential for CHIP’s neuroprotective role [117]. These findings helped conceptualize the idea that CHIP contributes to the triage of soluble polyQ proteins. Likewise, an HD transgenic mouse model revealed that the haplo-insufficiency of CHIP exacerbates disease pathology [117]. Similarly, CHIP overexpression led to increased degradation and ubiquitination of two common proteins that contain polyQ tracts, huntingtin, and ataxin-3, the main driver of SCA3 [118]. The age-dependent aggregation of polyQ-expanded ataxin-3 observed in SCA3 mouse models accelerates in the presence of CHIP haploinsufficiency, reinforcing the protective role of CHIP in polyQ diseases [119]. 

Recent studies also point to a neuroprotective role of CHIP in the clearance of polyQ aggregates. Mass spectrometry data of ataxin-3 interacting partners showed enrichment of CHIP interaction with ataxin-3 82Q compared to wild-type ataxin-3 [120]. Furthermore, the degradation of ataxin-3 82Q required both TPR and U-box domains of CHIP, indicating chaperone-mediated substrate recognition and subsequent ubiquitination [120]. In U2OS cells, treatment with YM-1, an allosteric activator of HSP70, reduced the mutant huntingtin aggregation and nuclear accumulation [121]. Furthermore, they proposed a model in which YM-1 increases the affinity of HSP70 to client proteins by stabilizing HSP70 in an ADP-bound state, which provides CHIP with sufficient time to ubiquitinate mutant huntingtin protein and promote its degradation [121]. Together, these data highlight CHIP and the UPS as potential therapeutic targets for treating polyQ diseases. 

### 2.4. Cancer

CHIP appears to function in opposing roles in cancer (Table 1, Figure 4). CHIP acts as a tumor suppressor in some cancer types, such as pancreatic cancer, breast cancer, and head and neck cancer [122,123,124]. CHIP expression is lower in these cancers than in healthy tissues, and low CHIP levels correlate with poor prognosis [122,123,124]. A similar tumor suppressor role for CHIP occurs in other cancer types, including lung, renal, and prostate cancer [125,126,127,128,129]. In the pancreatic cancer cell line BxPC-3, CHIP ubiquitinates and promotes the degradation of epidermal growth factor receptor (EGFR), and overexpression of CHIP suppresses cell growth [122] and is consistent with a tumor-suppressive role for CHIP in glioblastoma [130]. In breast cancer, CHIP targets human epidermal growth factor receptor 1 (Her2)/ErbB2, a member of the EGFR family, for degradation [131]. Given that Her2 is a promising target for developing inhibitors to prevent breast cancer growth, increasing CHIP expression or activity could suppress tumor growth via reducing Her2 receptors [132]. CHIP also regulates ovarian tumor domain-containing protein 3 (OTUD3), a deubiquitinase that stabilizes phosphatase and tensin homolog (PTEN), a frequently mutated tumor suppressor that plays a role in tumorigenesis [129,133]. Von–Hippel–Lindau (VHL) is a component of a multimeric protein complex that functions as a ubiquitin ligase. *VHL* is commonly mutated in renal cancer, and the subsequent loss of VHL-dependent ubiquitin ligase activity contributes to tumor growth and metastasis [134]. In renal cancer, CHIP targets transglutaminase 2 (TG2), a negative regulator of VHL [134]. Through this pathway, CHIP suppresses renal cancer proliferation via ubiquitination and degradation of TG2 [134]. CHIP also functions in a tumor suppressor role by inhibiting prostate cancer cell proliferation [128]. Furthermore, in head and neck cancer, CHIP overexpression reduces the proliferation, colony formation, and migration of HN13 and UMSCC12 cell lines, whereas CHIP knockdown results in increased tumor growth and cancer cell proliferation [124].

CHIP functions as an oncogenic protein in some cancer systems by mediating ubiquitin-proteasome-dependent degradation of tumor suppressor genes (Figure 5). Most notably, CHIP promotes the degradation of tumor suppressor proteins such as p53 and PTEN [135,136]. CHIP contributes to radiotherapy resistance in lung cancer by ubiquitinating and degrading p21, a CDK inhibitor; likewise, CHIP knockdown sensitized lung cancer cells to radiotherapy [137]. Furthermore, in prostate cancer models, CHIP can activate the Akt pathway, and overexpression of CHIP results in increased cell proliferation [138]. In colorectal cancer, CHIP functions as an oncogene by activating MAPK and AKT signaling pathways, resulting in increased cancer cell proliferation and migration [139]. Following the same trend, CHIP overexpression increased proliferation and colony formation in U251 and U87 glioma cell lines [140]. Remarkably, in some cancer types such as prostate and glioma, CHIP can act as both an oncogenic protein and a tumor suppressor; therefore, further investigation is required to elucidate the CHIP targets and function that determine CHIP’s role in cancer progression [128,130,138,140].

**Table 1 cells-10-03121-t001:** CHIP function in different cancers.

Cancer	CHIP’s Role	Target	Reference
Breast Cancer	TS	HER2/ ErbB2	[131]
Ovarian cancer	TS	OTUD3	[129,133]
Renal cancer	TS	TG2	[134]
Head and Neck cancer	TS	unknown	[124]
Lung cancer	OG	p21	[137]
Colorectal Cancer	OG	MAPK and AKT	[139]
Prostate cancer	TS	unknown	[128]
	OG	Akt	[138]
Glioblastoma/Glioma	TS	EGFR	[122,130]
	OG	unknown	[140]

TS = tumor suppressor, OG = oncogene.

Outcome data related to CHIP expression also point to a dichotomous role for CHIP and cancer. In gallbladder cancer, increased levels of CHIP expression are associated with a worse prognosis after surgery, whereas in pancreatic cancer and breast cancer, higher levels of CHIP expression are associated with higher survival [122,123,141]. Future studies focused on chaperone networks, and non-canonical functions of CHIP will hopefully reveal therapeutic targets related to CHIP and cancer.

### 2.5. Autoimmune Diseases

Regulatory T cells (Tregs) function as the break of the immune system. Tregs maintain immune/inflammatory homeostasis by suppressing the immune response via inhibiting cytokine release and T cell proliferation. When Treg function is compromised, the loss of control over immune activation can lead to autoimmune disorders [142]. Multiple studies identified how CHIP suppresses Treg function, suggesting that aberrant CHIP regulation could contribute to autoimmune diseases. Chen et al. demonstrated how CHIP blocked the immuno-suppressive role of Tregs by promoting the ubiquitination and degradation of Foxp3, a transcription factor that supports Treg function [143]. Follow-up studies from the same group showed the active derivative of vitamin A (all-trans RA) stabilized Tregs via downregulating CHIP expression, relieving Foxp3 inhibition [144]. A study on the mechanism of the antihistamine cimetidine also supports the CHIP-Foxp3-Treg mechanism. Cimetidine suppressed Treg function by CHIP-mediated Foxp3 degradation [145]. In addition, CHIP was upregulated in systemic lupus erythematosus patients’ CD4^+^ T cells and resulted in the ubiquitination and degradation of regulatory factor X 1 (RFX1), a transcription factor that suppresses systemic lupus erythematosus [146]. 

## 3. BAG Family Proteins 

There are six BAG (Bcl-2-associated athanogene) proteins, evolutionarily conserved throughout different species and named for the presence of at least one 50 amino acid BAG domain [147,148,149]. BAG proteins bind to the ATPase domain of HSP70 family chaperones and serve as a nucleotide exchange factor [150,151,152]. BAG proteins contain additional domains that mediate protein–protein interactions, including single polyproline (PxxP) regions, WW domains, and ubiquitin-like domains. Via HSP70 and BAG-dependent protein interactions, BAG family proteins regulate multiple cellular processes such as differentiation, division, apoptosis, and migration [151]. 

### 3.1. Heart Disease

BAG3 is expressed in the heart and regulates the ATPase activity of HSP70 family chaperones, including HSC70 and HSP70 [153]. These two members of the HSP70 family share 85% sequence similarity [154] but differ in their expression patterns; while HSC70 is constitutively active in the heart, HSP70 is expressed in response to various stressors [7]. BAG3 mutations cause a heritable form of dilated cardiomyopathy (DCM, Figure 3). DCM comprises 30%–40% of all HF cases and is one of the leading causes of sudden heart death [155] and heart transplantation [156]. The primary pathophysiology of DCM is the dilation and enlargement of one or both ventricles, with less than 40% left ventricular ejection fraction, which indicates inadequate ventricular contractility [155,157]. Interestingly, sex plays a role in the prognosis of DCM patients with BAG3 mutations; females had a better prognosis and developed fewer cardiac events than their male counterparts [158]. Furthermore, reduced expression of BAG3 in myofilaments occurred only in male patients [159]. 

Increased BAG3 expression occurs during both physiological hypertrophy and pathological remodeling, processes that can preserve healthy heart function or worsen it, respectively [160]. These observations highlight the importance of BAG3 in regulating cardiomyocyte responsiveness to stimuli. Likewise, genetically manipulating BAG3 in mouse models also perturbs cardiac function. BAG3 haploinsufficiency in mouse models increased apoptosis in the heart, increased heart size, and reduced left ventricular ejection fraction [161]. Human BAG3 overexpression in mouse models reduces fractional shortening, indicating a deteriorating heart condition [162].

Furthermore, BAG3 overexpression in the heart muscle of a CryAB R120G Tg mouse model reduces the fractional shortening and promotes the release of atrial natriuretic peptide (ANP), a physiological response to low blood pressure [163]. Heart-specific BAG3 loss-of-function in an αMHC-Cre mouse model increases ANP release, heart size, and fibrosis, which are the indicators of coronary heart disease [164]. Partial loss of BAG3 diminishes the contractility of human CMs [165]. Interestingly, modifying the endogenous BAG3 in mice to mimic the human BAG3-P209L mutation does not induce cardiomyopathy in transgenic knock-in mice up to 16 months of age [166]. However, when the human form is overexpressed with cardiomyocyte-specific αMHC P209L BAG3 Tg mice, gradually they develop HF by one year of age even though no observed HF indicators were present at birth [167]. 

*BAG3* is regulated transcriptionally by heat shock factor 1 (HSF1), an HSP70 and stress-induced transcription factor that regulates numerous genes involved in protein triage [168]. In healthy human iPSC-derived cardiomyocytes edited with the CRISPR-Cas9 system, a heterozygous knock-in DCM-associated mutation, BAG3-R477H, and a BAG3 knockout decreased BAG3/HSP70 complex formation and resulted in myofibrillar disarray under proteasome inhibition [169]. Induction of the heat shock response by lentiviral HSF1 transduction in heterozygous BAG3-R477H IPSC-derived CMs alleviated the proteasome inhibition-induced myofibrillar disarray compared to the controls, indicating the potential therapeutic effects of HSF1 in BAG3-associated DCM [169]. However, it is unclear if the therapeutic effect of HSF1 is dependent only on *BAG3* transcriptional induction or other HSF1 target genes involved in proteotoxic stress responses [169]. 

### 3.2. Cancer 

BAG1-L is the largest isoform of BAG1 and is the only isoform that contains a nuclear localization sequence, allowing it to function in the nucleus and regulate nuclear hormone receptors, including the androgen receptor (AR) [170,171]. BAG1-L inhibition is a promising treatment of AR-dependent prostate cancer as BAG1-L knockdown decreased cancer cell proliferation by reducing AR signaling [172,173]. Furthermore, BAG1-L is used as a biomarker for the prognosis of breast cancer (Figure 4) [174,175]. BAG2 accelerates the ATPase cycle on HSP70 and can change the refolding and degradation rates of HSP70 client proteins [150]. Interestingly, BAG2 functions as an oncogene in multiple cancer types (Figure 4). In esophageal carcinoma, oral cancer, and gastric cancer, BAG2 overexpression promotes cancer cell proliferation and is associated with poor prognosis [176,177,178]. BAG2 activates the MAPK pathway and ERK1/2 signaling in oral cancer and gastric cancer, respectively, as seen with BAG2 overexpression [177,178]. BAG2 also modulates estrogen receptor signaling by inhibiting CHIP expression and promoting the overexpression of mouse double minute 2 homolog (MDM2), an estrogen receptor modulator [116]. Additionally, BAG2 can induce pro-apoptotic pathways in response to proteasome inhibition in thyroid carcinoma cells [179]. 

BAG3 is another important BAG-family protein member in cancer progression (Figure 4). Increased BAG3 expression is shared across all cancer types and can create a desirable microenvironment for cancer progression in pancreatic ductal adenocarcinoma, melanomas, lung cancer, breast cancer, and prostate cancer [180]. Since there is a broad involvement for BAG3 in multiple different cancer types, research has focused on BAG3 inhibitors [181]. A BAG3 inhibitor showed promising efficacy in inhibiting cancer cell proliferation in breast cancer, prostate cancer, pancreatic cancer, and lung cancer cell lines [182]. Interestingly, Rosati et al. found that serum from pancreatic cancer patients contained BAG3, and pancreatic ductal adenocarcinoma cells secrete BAG3 [183]. Therefore, other therapeutic approaches that target BAG3 include the use of neutralizing antibodies. Showing the promising results of these approaches, anti-BAG3 and anti-PD1 treatment with targeted antibodies in a mouse model reduced pancreatic tumor volume along with an increase in CD8+ T cells [184]. Antibodies against BAG3 also inhibited the growth of pancreatic cell xenografts [185]. One confounding factor in developing anti-BAG3 therapeutics for cancer is that BAG3 plays an essential role in proper heart function, as discussed above. Therefore, BAG3 inhibitors should be extensively tested for their effects on heart function, and localized delivery methods for the inhibitors should be investigated. 

## 4. HOP/Stress-Inducible Phosphoprotein 1 

HOP, first discovered as a heat-inducible gene in yeast, is an HSP90 and HSP70 co-chaperone that facilitates client refolding [186,187]. HOP contains three TPR domains and can simultaneously bind to HSP90 and HSP70, enabling client transfer between HSP90 and HSP70 [188,189,190,191,192,193,194,195]. Furthermore, the interaction between HOP and HSP90 stabilizes HSP90 in open conformation which leads to non-competitive inhibition of HSP90 ATPase activity [190,196,197,198]. The importance of HOP in protein quality control was highlighted in a recent paper by Bhattacharya et al. Remarkably, HOP expression is necessary for proper proteasome assembly and, remarkably, HOP knockout cells effectively compensate for the impaired UPS via a compensatory increase in protein refolding [199]. A more detailed explanation of interactions between HOP and HSP90/HSP70 is covered extensively in another review [200]. STI1 is the predominant name used in neurodegeneration studies, whereas in cancer, HOP is commonly used.

### 4.1. Neurodegeneration

#### 4.1.1. Alzheimer’s Disease 

Neurons secrete STI1 that subsequently binds to cellular prion protein (PrP^c^) [201]. The binding between STI1 and PrP^c^ is neuroprotective, resulting in neuritogenesis and neuronal growth and survival [202,203,204,205,206,207]. For example, STI1-PrP^c^ interaction prevents amyloid-b oligomer (AbO) induced toxicity by inhibiting PrP^c^ binding to AbO in cell culture models and primary mouse neurons; this protective effect depends on ternary complex formation with HSP90, STI1, and PrP^c^ [208,209]. Interestingly, STI1 levels increase in the hippocampus of AD mouse models and the human AD cortex (Figure 2) [208]. In addition to STI1’s effects on AbO toxicity, STI1 loss-of-function increases tau toxicity in the fly retina [210]. However, no other follow-up studies investigated the role of STI1 in tau protein regulation. A recent study challenged the neuroprotective role of STI1 in AD mouse models. Lackie et al. showed that overexpression of STI1 in the 5xFAD mouse model exacerbated the AbO burden and increased memory deficits [211]. 

#### 4.1.2. Parkinson’s Disease 

HSP90 mediates α-synuclein aggregation in an ATP-dependent manner; therefore, co-chaperones that inhibit HSP90 ATPase activity, such as STI1, appear to prevent HSP90-dependent α-synuclein aggregation (Figure 2) [212]. Along the lines of this idea, a study showed that STI1 delayed PD-associated mutant α-synuclein-A53T accumulation [213]. 

#### 4.1.3. Huntington’s Disease and Amyotrophic Lateral Sclerosis (ALS)

In HD, loss of STI1 worsens PolyQ-htt-induced toxicity while the increase in STI1 is protective against it (Figure 2) [214,215]. Paradoxically, a genetic screen to identify mediators that regulate mutant Huntington identified knocking down the STI1 homolog in Drosophila reduced the proteotoxicity [216]. In the TDP-43 yeast model for ALS, STI1 deletion resulted in increased TDP-43 toxicity [217]. Interestingly, while moderate overexpression of STI1 protected against TDP-43 toxicity, high levels of STI1 exacerbated it [217]. It is clear that in HD and ALS, proteotoxicity is sensitive to levels of STI1 expression, and additional studies that include other components of the chaperone/co-chaperone machinery may lead to important insights into disease mechanisms. 

### 4.2. Cancer

Given the ability of HOP to facilitate protein folding by coordinating with HSP70 and HSP90, increased expression of HOP in the backdrop of cancer creates a pro-folding environment facilitating the folding and accumulation of oncogenic proteins, such as HER2, Bcr-Abl, c-MET, and v-Src (Figure 5) [218,219,220]. For example, HOP expression and HOP–HSP complex formation were higher in colonic carcinoma than non-tumor tissue samples [219]. The mRNA encoding HOP and HOP protein levels were more elevated in gastric tumor samples than non-tumor tissue samples, suggesting HOP expression may be an effective predictor of gastric cancer-related mortality [221]. While increasing HOP expression is associated with a pro-cancer phenotype, reducing HOP expression was sufficient to decrease cell proliferation, migration, and invasion in osteosarcoma and colorectal cancer cell models (Figure 4) [222,223,224].

In addition to the chaperoning of oncogenic proteins, HOP also contributes to a pro-cancer cellular environment by regulating signaling proteins that are substrates of HSP70 and HSP90. In turn, HOP’s co-chaperone activity can activate several signaling pathways, including JAK/STAT, AKT, and MAPK. For example, Hop maintains the stability of JAK2, the upstream regulator of STAT3, contributing to tumor growth and metastasis in both melanoma and ovarian cancer [225,226]. In colorectal cancer tissue samples, HOP expression correlated with STAT3 signaling, poor survival, and advancing stages of cancer, whereas HOP knockdown in colorectal cancer cells reduced proliferation, invasion, and migration [223,224]. Supplementing growth media with recombinant HOP protein stimulated proliferation in glioma cells and effect dependent on activation of TRAP1/AKT and MAPK/PI3K signaling [227]. In complementary experiments, knockdown of HOP reduced glioma cell proliferation and increased apoptosis; additionally, analysis of 153 glioblastoma patient samples revealed a positive correlation between HOP and TRAP1 expression [228].

These data highlight HOP as a promising molecular target for cancer therapies. Suppression of HOP activity or expression renders tumor cells susceptible to the stress of rapid proliferation, ultimately slowing tumor growth. Currently, there are no small molecules that directly inhibit HOP [229]. However, HOP/HSP90 complex inhibitors block HSP90 binding to HOP, resulting in anti-cancer effects [230,231,232,233,234]. Additional therapeutic targets include the post-translational modifications of HSP70, HSP90, and HOP, as these modifications impact the HOP/HSP interaction. Acetylation and phosphorylation of HSP70 increase the affinity for HOP binding over other co-chaperones, including CHIP (Figure 5) [64,67]. A similar phosphorylation site in HSP90 also promotes HOP versus CHIP binding [64]. In proof-of-principal experiments, blocking HSP90 acetylation reduced the HOP-HSP90 interaction and inhibited cancer growth [235]. Conversely, phosphorylation of HOP inhibits binding to heat shock proteins and decreases substrate refolding [195] offering another possibility to target HOP for cancer therapies. Recently, HOP was found to have intrinsic ATPase activity, which opens the possibility of small molecule inhibition targeting this domain [236].

## 5. FKBP51 & FKBP52

FKBP51 and FKBP52 are members of the peptidyl-prolyl cis-trans isomerase (PPIase) family, identified alongside HSP90 as a part of steroid hormone complexes [237,238,239,240,241]. FKBP51 and FKBP52 bind to HSP90 via their TPR domains and regulate steroid hormone receptors independent of FKBP51/FKBP52’s PPIase activity [242,243,244,245]. FKBP51 and FKBP52 regulate multiple signaling pathways, and we point the reader to several in-depth reviews on the biological functions of FKBP51 and FKBP52 [246,247,248]. 

### 5.1. Alzheimer’s Disease

Overexpression of FKBP51 in HeLa cells prevented tau degradation, resulting in increased total tau and phosphorylated tau levels (Figure 2) [249]. Furthermore, FKBP51 interacted with tau in the axonal tracts and promoted microtubule polymerization in a tau-dependent manner in vitro [249] and subsequently confirmed in vivo [250]. Blair et al. showed that tau levels were reduced in FKBP51 knockout mouse brains, whereas overexpression of FKBP51 in the tau transgenic mouse model resulted in the accumulation of tau oligomers [250]. They also identified an age-dependent increase in FKBP51 in healthy human brains with even higher levels of FKPB51 measured in AD brains [250].

Although FKBP52 shares 75% sequence similarity with FKBP51, their tau pathology roles appear different [247]. FKBP52 expression is reduced in the frontal cortex of human AD and FTDP-17 brain samples [251]. FKBP52 binds to hyperphosphorylated tau in the distal part of the axons and prevents tau-mediated microtubule assembly [252]. The same study also showed that FKBP52 overexpression decreased tau accumulation in differentiated PC12 cells [252]. In HeLa cells, knocking down FKBP52 increased total tau levels [253]. FKBP52 also interacts with a pathological tau mutant (P301L), and knockdown of FKBP52 in the transgenic tau-P301L zebrafish model rescued the associated axonal growth and branching defects with tau-P301L [254]. Furthermore, the truncated form of tau and caspase cleaved tau species can bind to FKBP52, and these interactions promote tau oligomerization and aggregation [255,256]. Interestingly, the interactions between FKBP52 and different tau species are independent of FKBP52’s PPIase activity [257]. New studies investigated the role of FKBP52 in tau pathology and tau-mediated cognitive deficits in wild-type and tau transgenic mouse models [258,259]. FKBP52 overexpression in aged wild-type mice resulted in increased phosphorylation of AD-associated tau species and impairments in spatial reversal learning [258]. Contrary to this observation, FKBP52 overexpression in rTg4510 mice failed to show an increase in phosphorylated tau species [259]. However, they observed a decline in spatial learning and increased neuronal loss in the hippocampus of rTg4510 mice overexpressing FKBP52, further highlighting the beneficial effects of FKBP52 inhibition in AD [259]. 

### 5.2. Cancer

FKBP51 and FKBP52 are linked to hormone-dependent cancers such as ERα-dependent breast and AR-dependent prostate cancer (Figure 4) [260]. For example, the expression of FKBP51 and FKBP52 was higher in both breast cancer and prostate cancer tissues compared to normal tissues [261,262]. In prostate cancer, FKBP51 and FKBP52 promoted cell proliferation by regulating AR’s nuclear translocation and dimerization [263]. In a complementary study, knocking down FKBP51 in the human prostate cancer cell line, LNCaP, decreased cancer cell proliferation along with decreased NF-kB signaling [264]. FKBP51 promotes the epithelial-to-mesenchymal transition through NF-kB signaling activation in papillary thyroid carcinoma cell lines K1 and TPC-1 [265]. However, additional experiments looking at the cytoskeleton formation to indicate increased migration and invasion were not seen with FKBP51 overexpression [265].

In contrast, rather than promoting tumorigenesis, FKBP51 overexpression decreased the proliferation of endometrial adenocarcinoma cell lines by inhibiting the Akt signaling pathway. In pancreatic cancer, FKBP51 also acts as a tumor suppressor by negatively regulating Akt phosphorylation [266]. Decreasing FKBP51 expression resulted in increased Akt phosphorylation and cancer growth, measured via cell proliferation in the SU86 cell line proliferation and tumor size in a mouse model [266,267]. Together, these studies suggest that FKBP51 affects unique signaling pathways depending on the cellular context, an important consideration in targeting FKBP51 for therapies.

### 5.3. Therapeutics 

FKBP51 and FKBP52 are potential therapeutic targets for certain cancers and AD. For example, FKBP51 and FKBP52 inhibitors decreased AR-dependent prostate cancer cell proliferation [260,263,268]. However, FKBP51 and FKBP52 inhibitor selectivity is a concern as the PPIase domains of FKBP51 and FKBP52 share high similarities with other FKBP proteins, and current PPIase inhibitors fail to show selectivity [247]. Whereas PPIase inhibition may be therapeutically effective in cancer, in AD models, there are PPIase-independent activities of FKBP51 and FKBP52 that appear to be important [245,257]. To that end, blocking the interaction between FKBP51 and FKBP52 to HSP90 is another strategy for therapeutics. However, these inhibitors likely block interactions between other TPR domain-containing proteins and HSP90 [246,247], again complicating the specificity of this approach. Recent efforts to identify selective inhibitors for FKBP51 and FKBP52 focus on using molecular dynamics simulations to achieve isoform selectivity [269]. 

## 6. CryAB

### 6.1. Cardioprotection 

CryAB (Alpha-crystallin B chain), classified initially as a chaperone, also functions as a co-chaperone [270]. Desmin is a crucial intermediate filament in cardiac muscles [270]. As of now, there are no adequate therapies for Desmin-related cardiomyopathies (DCRM) [271]. CryAB mutation R120G is a missense mutation that brings about a severe form of DRCM riddled with the accumulation of misfolded proteins (Figure 3) [271]. Using transgenic mice that overexpress CryAB, researchers demonstrated how over-expressing CryAB R120G mutant in CMs triggered aggregate accumulation intracellularly eventual heart failure by 12 months of age [271]. Moreover, mutation-driven disruptions in the CryAB/desmin interaction results in myofibrillar disarray, protein aggregation, heart dysfunction, and abrupt cardiac death [271,272,273]. In failing human hearts, Bouvet et al. found increased insoluble CryAB, soluble desmin, and hyperphosphorylated desmin levels [270]. Hyperphosphorylation of desmin leads to its aggregation, disrupting the cardiac muscle cytoskeleton and ultimately results in cardiomyopathy [274]. As a co-chaperone of HSC70, CryAB clears hyperphosphorylated desmin aggregates formed during ischemic HF via chaperone-assisted selective autophagy [270]. Another study identified insoluble aggregates positive for CryAB in murine cardiomyocytes [163]. Mutant CryAB-R120G mice have reduced heart contractility, a rise in insoluble aggregates in CMs, and an increase in BAG3 compared to control mice by eight months of age [163]. Overexpressing BAG3 in the heart muscle of a CryAB-R120G Tg mouse resulted in BAG3-mediated CryAB degradation [163]. Another study identified the BAG3-P209L mutation in a pediatric heart with left ventricular wall thickening and larger atria [275]. Interestingly, IHC staining showed CryAB, desmin, and ubiquitin present in the intracytoplasmic inclusions with a slight increase in overall desmin levels in the mutant heart compared to control tissue [275]. 

### 6.2. Cancer 

CryAB is an essential factor in multiple cancers (Figure 4). CryAB inhibits migration and invasion of the cancer cells in bladder cancer cells by decreasing PI3K and AKT signaling, suggesting that CryAB acts as a tumor suppressor [276]. Paradoxically, in gastric cancer tissue samples, CryAB expression was higher than normal tissues [277]. Similarly, in colorectal cancer cell lines, CryAB expression is upregulated and promotes metastasis and invasion [278]. Furthermore, inhibiting CryAB in vitro induced cancer cell apoptosis and in vivo decreased migration and tumorigenesis [278]. CryAB may play an oncogenic role in osteosarcoma, as the downregulation of CryAB decreased cancer cell proliferation [279]. Collectively, these studies highlight CryAB as a potential therapeutic to decrease tumor progression.

### 6.3. Multiple Sclerosis

CryAB plays an essential role in multiple sclerosis, an autoimmune disease that results in the demyelination of the central nervous system. In addition to being the most prominent mRNA expressed in early multiple sclerosis lesions [280], the accumulation of CryAB at these lesions is reversible [281]. Clinical trials using CryAB immunotherapy slowed disease progression in multiple sclerosis patients [280,282,283,284]. However, it is unclear if CryAB is working through its autonomous chaperone function or via HSP/co-chaperone function, highlighting a gap in our current knowledge base regarding the CryAB-MS mechanism. 

## 7. Sgt1 

Sgt1 was first identified in yeast as a regulator of SCF (Skp1-Cul1-F-box) E3 ubiquitin ligase complex, and later on, its expression was confirmed in the mammalian tissues such as the brain, liver, and lungs [285,286]. Sgt1 interacts with HSP90 through its CHORD domain, and together Sgt1-HSP90 ensures the proper kinetochore assembly [287,288,289,290]. Sgt1’s role as a co-chaperone was established by observing that Sgt1 levels are upregulated upon heat shock, and Sgt1 prevents aggregate formation in citrate synthase aggregation assay [291]. 

### Neurodegenerative Diseases

Although Sgt1’s role in neurodegeneration has not been extensively studied, a decrease in Sgt-1-immunopositive neurons in the cerebral cortex of the AD brain has been identified, indicating Sgt-1 might serve a neuroprotective role in AD (Figure 2) [292]. On the contrary, a recent study identified upregulation of *Sgt-1* mRNA levels in the temporal and frontal cortex of PD patients with no significant changes in the protein expression [293]. These data suggest that Sgt-1’s role in neurodegeneration might be disease-specific and requires further investigation. 

## 8. HSP40/DNAJ Protein Family

HSP40/DNAJ proteins are a class of molecular chaperones/co-chaperones that regulate protein translation, folding, unfolding, translocation, and degradation [294,295]. These roles are primarily carried out by forming a complex with HSP70 via a conserved J-domain to enhance the ATPase activity of HSP70 [294,295]. HSP40/DNAJ proteins, alongside with HSP70, interacts with disease-causing misfolded proteins and promotes their clearance [295,296,297,298,299]. Therefore, HSP70 and its co-chaperones, HSPsp40/DNAJ proteins, are considered potential therapeutic targets in cellular and animal models of ataxia and other neurodegenerative conditions [296].

### 8.1. DNAJC3

DNAJC3 is a co-chaperone of BiP (immunoglobulin heavy-chain-binding protein), an HSP70 family member. BiP, primarily residing in the endoplasmic reticulum, facilitates the folding of the nascent polypeptides and ensures homeostasis by mitigating the cellular stress response caused by unfolded proteins [297]. DNAJC3 assists BiP with the de novo folding of nascent proteins and targeting misfolded proteins for degradation [297,299]. Nonsense mutations in *DNAJC3* were identified in a consanguineous Turkish family in which three siblings were diagnosed with the autosomal recessive disorder ACPHD (ataxia, combined cerebellar and peripheral, with hearing loss and diabetes mellitus, Figure 2) [300]. Furthermore, they identified a loss-of-function mutation in a patient with diabetes who also presented hearing impairment and ataxia [300]. These findings indicate DNAJC3 mutations could be associated with ataxia phenotypes. 

### 8.2. DNAJC5

DNAJC5, also called cysteine string protein-a (CSPa), is a major presynaptic co-chaperone implicated in various neurodegenerative diseases [298]. DNAJC5 is primarily expressed in neurons to chaperone the synaptic SNARE protein SNAP-25 [298]. This chaperoning event facilitates the formation of the synaptic SNARE complexes that are vital for synaptic vesicle fusion to the plasma membrane for presynaptic neurotransmission [298]. DNAJC5 has various neuroprotective properties. Mutations in DNAJC5 cause autosomal dominant adult-onset neuronal ceroid lipofuscinosis (ANCL, Figure 2) [301,302]. ANCL, a neurodegenerative disease with symptoms like ataxia, seizures, and dementia, is characterized by the accumulation of lipofuscin, an autofluorescent lysosomal waste [302,303]. DNAJC5 KO mice showed deficiency in neuromuscular function and impairments in synaptic transmission, indicating that DNAJC5 expression is vital for proper synapse function [304]. 

To date, there are no other diseases that are associated with DNAJC5 mutations. However, alterations in the levels and activity of DNAJC5 impact other neurodegenerative conditions. DNAJC5 can interact with mutant huntingtin that contains an expanded PolyQ domain but not with the wild-type protein [305]. A proteomic study of the mouse brain also confirmed the association between DNAJC5 and mutant huntingtin protein [306]. However, DNAJC5’s role in Huntington’s Disease has yet to be established. 

### 8.3. Rheumatoid Arthritis

Proteins in the HSP40/DNAJ family contribute to autoimmune diseases such as rheumatoid arthritis. Patients with rheumatoid arthritis had elevated levels of autoantibodies to HSP40 and DNAJ proteins; moreover, HSP40 inhibited proliferation of T cells from these patients, consistent with a regulatory role for HSP40 in the immune response [307,308,309,310]. Additionally, Koffeman et al. developed an immunotherapy for rheumatoid arthritis using a peptide fragment of DNAJP1, a DNAJ family member, highlighting the utility of HSP40/DNAJ family proteins as potential therapies for autoimmune diseases [311].

## 9. Targeting Co-Chaperones with Small Molecules for Therapies

There are multiple functional domains of HSP70 and HSP90 that serve as possible targets for small molecule inhibition [312,313,314]. Conversely, there are diseases where activation of HSP70 to mimic thermal and immune preconditioning may be beneficial to outcomes [36,315]. We encourage the reader to look at in-depth reviews regarding the current landscape of targeting HSP70/90 function in cardiovascular disease [316,317], cancer [37,38,318], neurodegeneration [4,315], and inflammation [318,319]. More recent approaches include targeting the interactions between HSP90 and co-chaperones to inhibit chaperone function [320]. Given the importance of the C-terminal tail of both HSP70 and HSP90 interacting with TPR-co-chaperones, such as CHIP and HOP (Figure 5), targeting the chaperone tails to modify chaperone function is an emerging concept [321].

### 9.1. CHIP

Despite the data demonstrating the knockdown of CHIP decreases cancer proliferation, such as in colorectal and lung cancer, the literature is scant regarding small molecule targeting of CHIP. Complications with small molecule targeting of CHIP include off-target effects given the diverging roles of CHIP across cancer types (Table 1) and the implication of altered CHIP function in other organ systems (Figure 2 and Figure 3). 

### 9.2. BAG1

Enthammer et al. isolated a thioflavin (Thio-2) that inhibited BAG1 interactions and decreased growth of BRAF-resistant breast cancer cell line MCF7 [322]. Cato et al. found that Thio-2 attenuated the BAG-1L/Androgen receptor interaction and decreased androgen receptor-dependent pancreatic cancer growth [172]. In combination with other breast cancer chemotherapies, BAG1 down-regulation improved the effectiveness and cytotoxicity of the drugs in drug-resistant breast cancer cell lines [323]. Therefore, more research on BAG-1L inhibitors would be beneficial for decreasing tumor progression. 

### 9.3. BAG2

Despite the evidence that inhibition of BAG2 decreased cancer cell proliferation, as highlighted above in oral and gastric cancer, there are no reports of small molecule inhibitors of BAG2.

### 9.4. BAG3

One confounding factor in developing anti-BAG3 therapeutics for cancer is that BAG3 plays an essential role in proper heart function, as discussed above. Most recently, Martin et al. highlighted the cardiomyocyte toxicity of the cancer therapeutic JG-18 that targets the BAG3-HSP70 interaction [159]. Therefore, much like CHIP, BAG3 inhibitors should be extensively tested for their effects on heart function, as well as localized delivery systems.

### 9.5. CryAB

As mentioned in Section 6.2, CryAB is a potential therapeutic target for some cancers such as osteosarcoma, gastric and colorectal cancer. Chen et al. identified a structure-based small molecule inhibitor for CryAB that decreased triple-negative breast cancer cell growth [324]. However, this was one of the few studies investigating CryAB’s therapeutic potential in cancer progression, highlighting a gap for future work. 

## 10. Conclusions

Undoubtedly, co-chaperones impart a range of control over HSPs, allowing for the fine-tuning of responses to cellular stress. It is also clear that, although co-chaperones are prime candidates for targeted therapies, there is still much to learn about crosstalk between co-chaperones and the resiliency of biological compensation. We hope, in the years to come, that more studies will elucidate the broader scope of the chaperone/co-chaperone network and provide hope for so many harmful human diseases. 

## Figures and Tables

**Figure 1 cells-10-03121-f001:**
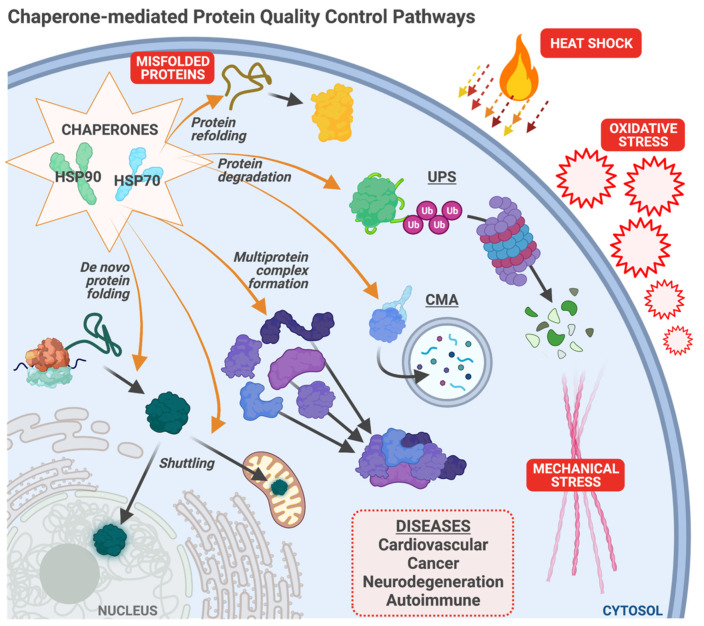
Chaperone-mediated protein quality control. Chaperones, including HSP70 and HSP90, maintain cellular homeostasis through multiple pathways: assisting with de novo protein folding; multiprotein complex formation; protein shuttling throughout the cell; degradation of terminally misfolded proteins (via the ubiquitin-proteasome system (UPS) and chaperone-mediated autophagy (CMA); and refolding of misfolded proteins damaged by cellular stress. The chaperone system responds to multiple stressors, including the accumulation of misfolded proteins, heat shock, oxidative stress, and mechanical stress. Chaperone dysfunction contributes to numerous diseases discussed in this review.

**Figure 2 cells-10-03121-f002:**
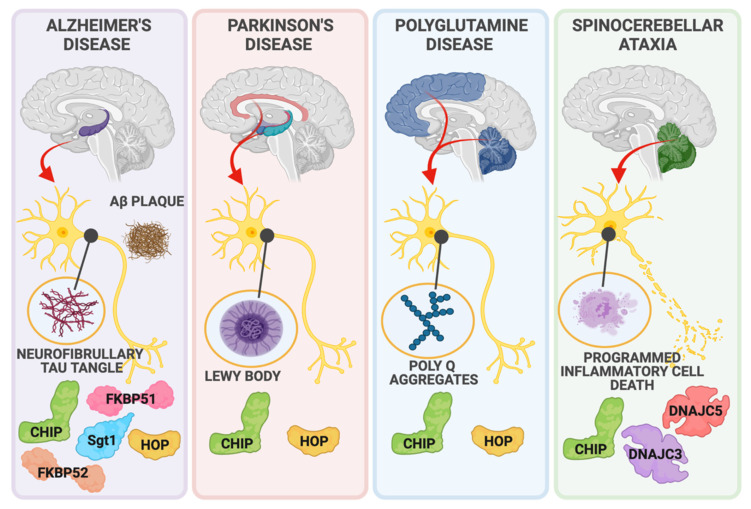
Co-chaperones and neurodegenerative disease. The dysfunction of numerous co-chaperones contributes to neurodegenerative disease pathologies found throughout the brain. HSP70 and HSP90 co-chaperones including CHIP, HOP, FKBP51, FKBP52, and STG1 interact with proteins and aggerates associated with Alzheimer’s disease (Tau, Amyloid-Beta), Parkinson’s (Alpha-synuclein, Lewy Bodies), and Polyglutamine disease (Poly Q aggregates). The co-chaperones DNAJC3, DNAJC5, and CHIP protect against neuronal death in spinocerebellar ataxias.

**Figure 3 cells-10-03121-f003:**
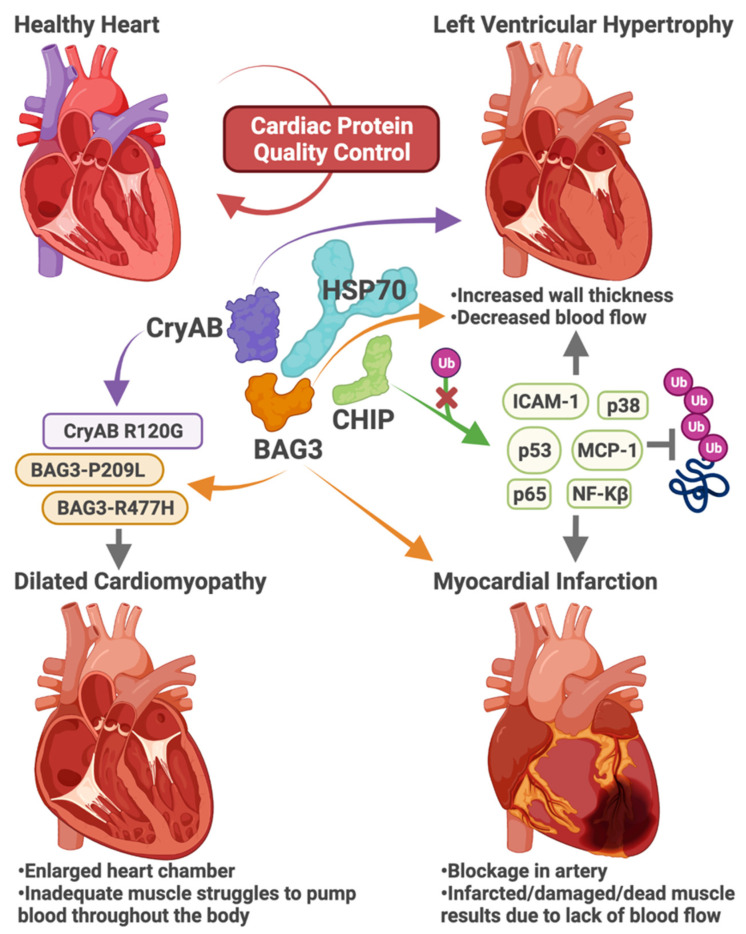
The role of co-chaperones in cardiac protein quality control. Heat shock proteins, including HSP70, coordinate with co-chaperones to maintain proteostasis in the heart. Impairment of cardiac protein quality control can lead to distinct forms of heart disease, including left ventricular hypertrophy, dilated cardiomyopathy, and myocardial infarction. Loss-of-function in the co-chaperone proteins CryAB, BAG3, and CHIP alters chaperone function and the ability to maintain proteostasis, leading to heart disease. Missense mutations in CryAB and BAG3 cause heritable forms of cardiomyopathies (purple and orange). Loss of CryAB or BAG3 function can lead to left ventricular hypertrophy or increased susceptibility to infarction, respectively. Finally, the ability of CHIP to ubiquitinate regulatory proteins in cardiomyocytes (green) is necessary to prevent cardiac hypertrophy and cell death in response to myocardial infarction.

**Figure 4 cells-10-03121-f004:**
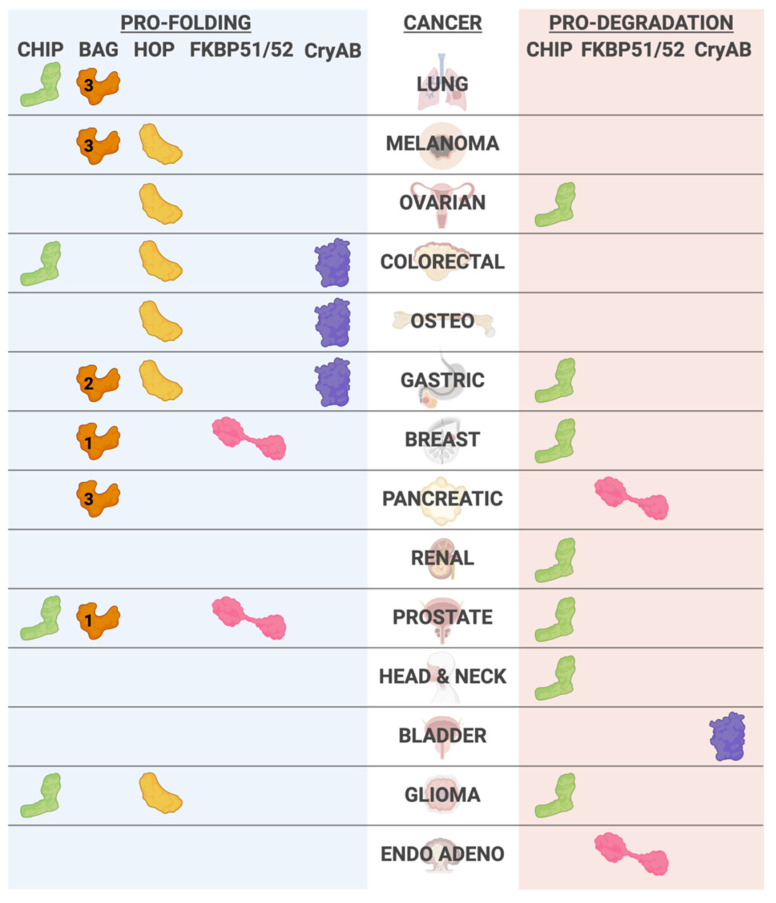
Co-chaperones and their implications across the cancer spectrum. Co-chaperones can associate with pro-folding and pro-degradation activities towards chaperone substrates. The co-chaperones listed indicate both the cellular triage condition, pro-folding (blue) or pro-degradation (orange), as well as the type of cancer (ENDO ADENO- endometrial adenocarcinoma, OSTEO- osteosarcoma). When appropriate, we included the specific BAG family member identifier; however, we did not indicate BAG2 associations with esophageal, oral, and thyroid cancer.

**Figure 5 cells-10-03121-f005:**
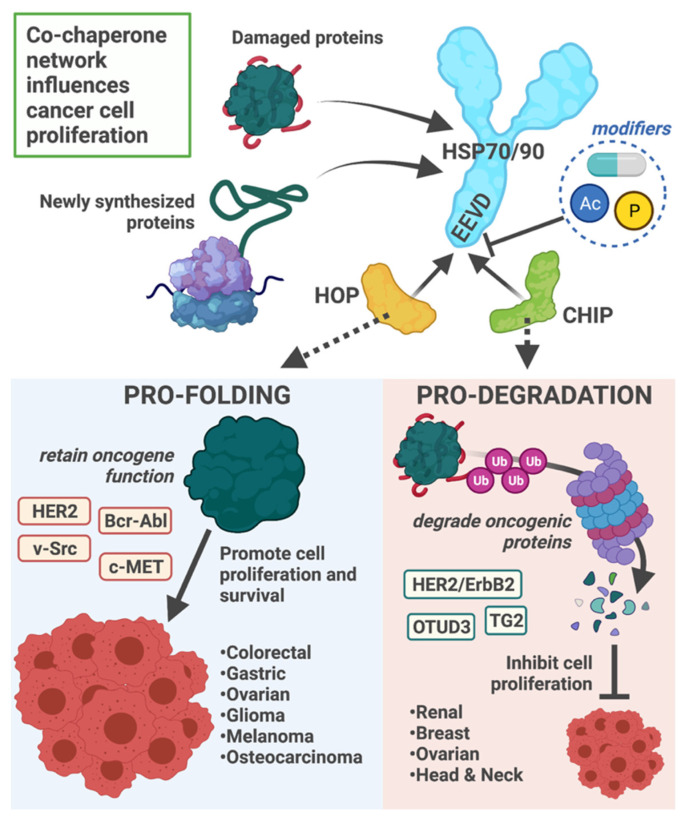
Co-chaperone network influences cancer cell proliferation. The decision to refold or degrade proteins represents an essential component of protein quality control. The co-chaperones HOP and CHIP compete for binding the EEVD motif located at the C-terminal tail of HSP70 and HSP90. The balance in HOP versus CHIP binding to HSPs results in a pro-folding or pro-degradation complex, respectively. In cancer, the pro-folding environment promotes cell proliferation by the constant re-folding of oncogenic proteins. In contrast, if CHIP–HSP binding is favored, oncogenic proteins can be degraded through the ubiquitin-proteasome system and ultimately inhibit cell proliferation. The cancers associated with these protein environments and identified substrates are provided. Additionally, the affinity of HOP and CHIP to HSPs are modified by post-translational modifications, including phosphorylation (P) and acetylation (Ac). Likewise, small molecules that target the C-terminus of HSP70 and HSP90 may influence co-chaperone occupancy. The C-terminal tail of HSP70/90 and post-translational modifications could be targeted to control the protein triage environment.

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
