# Peer review of "With or without You: Co-Chaperones Mediate Health and Disease by Modifying Chaperone Function and Protein Triage"

_cells, 2021, doi:10.3390/cells10113121_

Round 1

Reviewer 1 Report

This review is a descriptive narrative that looks at a selected number of diseases and how these are influenced or caused by co-chaperone mutation or changes in their levels or activity. There is extensive referencing throughout. I find it dismisses the potential of Hsp90 inhibitors, especially c-terminal binding modulators of Hsp90 activity, in being able to treat specific conditions. Perhaps a small section on this would conclude the section. What about drugs develop against co-chaperones. Is there anything here that has been developed and should be included in more detail.

The following points should be addressed as they are misleading or inaccurate:

1. I don't agree with the statement that Hsp90 inhibitors are not specific. In the context of a cancer cell, that is highly dependant on Hsp90, and can be up regulated to very high levels, its inhibition is significant to kill such cells. However, normal cells are able to up regulate Hsp90 to avoid any damaging effects. I think the statement is out of context and this should be addressed.

2. I do not agree with the statement 'Nucleotide exchange is the rate-limiting factor for chaperones to release their substrate,' within the CHIP section. The limiting factor for the chaperone cycle in Hsp90 is conformational change. This statement needs to reflect that or make it clear what it means.

Reviewer 2 Report

Very interesting review on Hsp and their co-chaperones related to myocardial, neuronal and cancer reasearch. Well structured paper with extensive well selected references.

Author Response

Thank you for the kind words!

Reviewer 3 Report

Altinok et al., have submitted the review article entitled “With or Without You: Co-chaperones mediate health and disease by modifying chaperone function and protein triage”.

The authors have made a substantial attempt to provide a comprehensive review covering cancer, neurodegenerative disease, and cardiovascular diseases.

This reviewer has two major suggestions.

  1. Despite very informative content, the authors should have covered the autoimmune disease as well. As many studies have been reporting the importance of chaperones in autoimmune diseases e.g., https://doi.org/10.1038/s41598-018-34887-6
  2. In addition, the authors should also provide the possible illustrations in the context of co-chaperones role in different diseases via modifying chaperone function. Illustrations would attract more readers.

Round 2

Reviewer 3 Report

The authors have revised the manuscript by considering reviewers suggestions. The present format of the manuscript looks interesting. Hereby I endorse the manuscript for publication.